# Effects of Machining Velocity on Ultra-Fine Grained Al 7075 Alloy Produced by Cryogenic Temperature Large Strain Extrusion Machining

**DOI:** 10.3390/ma12101656

**Published:** 2019-05-21

**Authors:** Xiaolong Yin, Haitao Chen, Wenjun Deng

**Affiliations:** School of Mechanical and Automotive Engineering, South China University of Technology, No. 381 Wushan Road, Tianhe District, Guangzhou 510641, China; 201610100078@mail.scut.edu.cn (X.Y.); 201720100128@mail.scut.edu.cn (H.C.)

**Keywords:** Al 7075 alloy, cryogenic, large strain extrusion machining, microstructure, micro-hardness

## Abstract

In this study, cryogenic temperature large strain extrusion machining (CT-LSEM) as a novel severe plastic deformation (SPD) method for producing ultra-fine grained (UFG) microstructure is investigated. Solution treated Al 7075 alloy was subjected to CT-LSEM, room temperature (RT) LSEM, as well as CT free machining (CT-FM) with different machining velocities to study their comparative effects. The microstructure evolution and mechanical properties were characterized by differential scanning calorimetry (DSC), scanning electron microscopy (SEM), transmission electron microscopy (TEM), and Vickers hardness measurements. It is observed that the hardness of the sample has increased from 105 HV to 169 HV and the chip can be fully extruded under CT-LSEM at the velocity of 5.4 m/min. The chip thickness and hardness decrease with velocity except for RT-LSEM at the machining velocity of 21.6 m/min, under which the precipitation hardening exceeds the softening effect. The constraining tool and processing temperature play a significant role in chip morphology. DSC analysis suggests that the LSEM process can accelerate the aging kinetics of the alloy. A higher dislocation density, which is due to the suppression of dynamic recovery, contributes to the CT-LSEM samples, resulting in greater hardness than the RT-LSEM samples.

## 1. Introduction

Ultra-fine grained (UFG) Al alloys and its preparation have raised increasingly attention recently due to its excellent mechanical properties compared with its coarse-grained counterparts. Generally, severe plastic deformation (SPD) techniques, including equal channel angular pressing (ECAP), accumulative roll bonding (ARB), high-pressure torsion (HPT), et al., are most commonly used methods to prepare UFG materials. Recently, an innovative SPD method called large strain extrusion machining (LSEM) was developed, which can impart large strains in a single step of deformation [1,2,3]. Deng et al. [4,5,6] utilized experimental methods in combination with finite element modeling (FEM) to study the influence of parameters such as rake angle, chip compression ratio and tool corner radius on the deformation behaviors and material properties in LSEM. Chandrasekar et al. [7] investigated the materials flow and deformation tensor fields in LSEM using high-speed imaging and PIV (particle image velocimetry), and they found that the deformation field is intense, narrowly confined and controllable. In order to suppress shear localization in LSEM, Rao et al. [8] suitably altered the texture of Ti-6Al-4V and eventually obtained fine-grained continuous Ti-6Al-4V foils. 

However, the effect of deformation temperature on the material properties and machinability during LSEM has been rarely studied. Large amounts of heat inevitably generated in machining will weaken the strength of materials due to the dynamic recovery [9]. Accordingly, eliminating the side effects of deformation temperature on the material is quite meaningful. In particular, there are formability restrictions for the Al alloys with high stacking fault energy at ambient temperature. Deformation at cryogenic temperature has been identified as a promising route to prepare UFG materials. Zhao et al. [10] presented a strategy to simultaneously enhance the ductility and strength of bulk nanostructured Al 7xxx alloys by cryo-rolling and subsequent low-temperature annealing. Tao et al. [11] investigated the microstructures and hardness of pure Al subjected to cryogenic dynamic plastic deformation (DPD), they concluded that the grain refinement is significantly benefited by increasing the strain rate and reducing the deformation temperature. In the research of Jayaganthan et al. [12], the hardness of cryo-rolled and RT-rolled Al 7075 samples increased from 105 HV to 190 and 180 HV respectively, at a rolling strain of 2.3. Yu et al. [13] successfully fabricated Al/Ti/Al laminate sheets with no edge cracks using cryogenic roll bonding. Thus, a combination of SPD processing and a cryogenic technique has great potential to improve the mechanical properties of various metals and alloys. Based on this, we put forward cryogenic temperature large strain extrusion machining (CT-LSEM) to prepare UFG materials while overcoming the aforementioned limitations. The LSEM process could impart large strains to the materials in a single step of deformation. Comparing with the conventional SPD methods, LSEM is more efficient to fabricate UFG materials. 

Figure 1 shows the schematic of CT-LSEM, in which chip undergoes machining and extrusion at the same time. A constraining tool is applied across from the cutting tool, forming the extrusion channel. The chip formation can be considered as a plane-strain process, when *α*, *t_ch_*, *t_d_* and Vc represent the rake angle, chip thickness, un-deformed chip thickness, and machining velocity respectively. The shear strain imposed in the chip is calculated by
(1)$$\varepsilon =\lambda /\cos\alpha +1/\left(\lambda \cos\alpha \right)-2\tan\alpha$$
where λ is the chip compression ratio:(2)$$\lambda =t_{ch}/t_{d}$$

In this article, the solution treated (ST) Al 7075 was processed by CT-LSEM with different machining velocities. The Al 7075 was ST in order to obtain a supersaturated solid solution (SSS) due to its higher work hardening rate and better microstructure refinement. Room temperature large strain extrusion machining (RT-LSEM) and free machining (FM) under CT (CT-FM) were also conducted for comparison. The machining velocity plays a very significant role in the chip formation process and is the main factor that influences the heat generation in the deformation zones. As the heat induced by deformation and friction is detrimental to the mechanical properties owing to some microstructure changes within the material, it is therefore necessary to reduce the heat effects on the material during machining. Additionally, the chips prepared by different methods might present distinctive morphologies. As far as we know, CT-LSEM with the starting materials in the ST state under various machining velocities has not been reported for precipitation hardenable alloy (Al 7075) in opened literatures. Since the strengthening mechanisms of 7xxx series alloy mainly include grain refinement, dislocation, and precipitation, their contributions to alloys subjected to RT and CT LSEM, as well as CT-FM are very worthy of research. Therefore, the overall purpose of present work mainly focuses on: (i) developing a new method to prepare UFG alloys; (ii) studying the effects of machining velocity on the chip formation; (iii) identifying the differences in microstructure evolution and mechanical properties; (iv) investigating the strengthening mechanisms of chip samples. The microstructural and mechanical features were examined by employing differential scanning calorimetry (DSC), scanning electron microscopy (SEM), transmission electron microscopy (TEM), and Vickers hardness measurements. 

## 2. Materials and Methods 

The commercial Al 7075 with the chemical composition of 5.6 Zn, 2.5 Mg, 1.6 Cu, 0.5 Fe, 0.4 Si, 0.3 Mn, 0.23 Cr, 0.2 Ti, and Al balance was served as the workpiece. Prior to experiments, the workpiece was solution treated (ST) at 490 °C for 6 h and then quenched in water in order to obtain a supersaturated solid solution. The ST alloy was then dipped into liquid nitrogen for at least 20 min to fulfill the cryogenic condition. Then, the alloy was immediately subjected to CT-LSEM. Subsequent CT-LSEM passes were performed after intermittent immersion of the alloy in liquid nitrogen for five minutes as shown in Figure 1. 

The annealed alloy without cryogenic treatment was also processed by RT-LSEM. Additionally, conventional machining under cryogenic condition, i.e. cryogenic temperature free machining (CT-FM) without the constraining tool was conducted for comparison. The experimental procedures and main parameters of CT-FM are in line with CT-LSEM. Chips were collected and kept in the fridge at a temperature lower than −20 °C to avoid natural aging. The α, $t_{d}$, *λ* were set 20°, 0.5 mm and 2 respectively, while the machining velocity Vc varied (5.4 m/min, 10.8 m/min, and 21.6 m/min). All machining experiments were conducted at a CA6140A lath. The experimental setup and associated parameters of LSEM are shown in Figure 2. 

The microstructure features of the Al 7075 alloys were characterized using a Nova Nano 430 scanning electron microscope (SEM) (FEI, Hillsboro, OR, USA) operating at 10 kV and a FEI Tecani F20 transmission electron microscopy (TEM) (FEI, Hillsboro, OR, USA) operating at 200 kV. SEM samples were mechanically polished to mirror finish and then etched with Keller’s reagent (95 mL H_2_O, 2.5 mL HNO_3_, 1.5 mL HCl, and 1.0 mL HF). As for TEM study, samples were first grounded to ~50 μm thickness followed by twin jet electron polishing with a solution of 10% perchloric acid and 90% ethanol at −25 °C and 20 V. DSC analysis was performed using a NETZSCH STA 449 C (NETZSCH, Selb, Bavaria, Germany) under pure argon atmosphere at a constant heating rate of 30 °C/min. Vickers hardness tests were measured on the plane parallel to the tool rake face by applying a load of 50 g for 15 s. For each sample, at least 15 readings were calculated to obtain the average value of hardness. 

## 3. Results and Discussion

### 3.1. Chip Thickness and Morphology

The chip formation process, which is mainly determined by combined effects including material properties and machining parameters, is especially complicated for LSEM. Unlike for conventional machining, the material undergoes extrusion and shearing simultaneously after it passes through the primary deformation zone, forming a geometrically controlled chip. Therefore, various machining conditions would result in different chip thickness and morphologies. Figure 3 shows the chip thickness variations versus machining velocities under RT-LSEM, CT-LSEM, and CT-FM conditions. The mean value of chip thickness for each condition was calculated from at least 20 measurements. In the case of chips produced by LSEM, it is interesting to note that the chip thickness *t_ch_* reduces slowly when the velocity increases from 5.4 m/min to 10.8 m/min and decreases significantly as the velocity increases to 21.6 m/min. The experimental chip thickness at 21.6 m/min is far less than the predefined value of 1 mm, which means that the chip may not be extruded and the LSEM process deteriorates into free machining. There is not enough time for the material to deform into the chip as expected when the machining velocity is very high. Therefore, chip formation behavior is greatly weakened. Due to the lack of constraining tool, the chip thickness values of CT-FM are maintained at about 0.8 mm, which is far less than the predefined value. 

Another noticeable phenomenon is that the chip thickness for the RT-LSEM is less than that for the CT-LSEM at each velocity. Also, the chip morphologies between them are definitely different as shown in Figure 4. Unlike RT and CT LSEM, the chips prepared by CT-FM are curly, which is owing to the absence of restriction effect of materials. The chip surfaces of the RT-LSEM are of bad quality with macroscopically visible cracks spreading on them. However, cracks can hardly be found in the chips fabricated by CT-LSEM, that is, the chips are featured with better integrity. It can thus be concluded that the chip thickness and morphology is temperature dependent. The constraining tool also plays a major role in the chip morphology. It indicates that once the appropriate processing conditions have been selected, UFG chips with better integrity, which are more suitable for practical applications, can be obtained. These findings could also provide guidance for our future work on tools optimization and practical application of chip materials.

During RT-LSEM, Mg and Zn solutes within the Al 7075 are prone to translate to GP (Guinier-Preston) zones due to the heat induced by deformation, which are harmful to the formability of Al alloys [14,15]. This would not occur in the CT-LSEM process because Mg and Zn solutes cannot get sufficient energy to form GP zones at CT. In combination with low velocity and CT, the chip can be fully formed and extruded. In addition, the strain hardening capability of fcc (face-centered cubic) alloys increases and crack localization is suppressed when deformed at low temperatures [16,17]. The dislocation cross and slip mechanisms and dislocation annihilation are suppressed because of the limited dislocation mobility at CTs, leading to the reduction of dynamic recovery rate [9,18]. Consequently, deformation at CT promotes the accumulation of dislocations, due to which the ductility and work hardening capacity of alloys are enhanced. This can be used to explain why the chips produced by CT-LSEM is thicker than RT-LSEM. In brief, CT-LSEM at a machining velocity of 5.4 m/min is a nice option to obtain chips with adequate thickness and good quality. 

### 3.2. DSC Analysis 

DSC thermograms contain endothermic and exothermic peaks, representing dissolution and formation of phases respectively. The 7xxx series alloy is a precipitation hardenable alloy. It is generally accepted that the supersaturated solid solution of 7xxx alloys decompose in the following sequence:

Supersaturated solid solution → GP zones → η’ (MgZn_2_) → η (MgZn_2_)

DSC thermograms of ST, RT-LSEM, CT-LSEM and CT-FM samples are shown in Figure 5. Based on the precipitation sequence, the DSC curves can be analyzed as follows. The trends are quite similar for all DSC curves. A large endothermic peak (label 1) appearing in all samples are probably corresponding to either dissolution of GP zones or metastable η’ (MgZn_2_) phases. However, no reactions corresponding to the formation of GP zones can be detected in all samples. It is reported that GP zones are formed at low temperatures ranging from 20 °C–125 °C [19]. Therefore, it can be deduced that GP zones are expected to be formed prior to the DSC run. Also, peak 1 in the LSEM and CT-FM samples occurs much earlier than that in the ST sample, indicating that the transition temperature from GP zones to η’ phases is lower for the deformed materials. On the other hand, a large exothermic peak (label 2) including a shoulder on the left can be observed in the three deformed samples, which is probably a mix of metastable η’ and equilibrium η phase formation reactions. For the ST sample, the formation of η’ and η precipitates are separate and they are marked by label 2a and 2b respectively. The large exothermic peak can be mainly attributed to the release of stored energy during the recovery of the deformed alloys [20]. Based on the above mentioned analysis, it can be concluded that the alloys subjected to LSEM and FM have a lot of room for improvement in strength once they are annealed with proper temperatures. This is because plenty of strengthening phases such as η’ and η would precipitate out from the matrix, resulting a large enhancement of strength of the materials. Therefore, the combination of CT-LSEM and subsequent aging treatment will be a focus of the forthcoming works. 

In the range of temperature between 375 °C to 450 °C, an endothermic peak (label 4) is dominant in all DSC curves, which might be attributed to the dissolution of η (MgZn_2_) phases obtained by over aging. In front of the peak 4, an obvious exothermic peak (label 3) is in present in the LSEM samples, which is missing in the ST sample. It may be caused by the coarsening of the η precipitates or the static recrystallization occurring during the DSC run. It should be noted that all reaction peaks corresponding to the formation and dissolution of phases in the LSEM samples shift to the left as compared to those of the ST sample. Similarly, all reaction peaks in the CT-FM and CT-LSEM samples shift a little to the left compared with that in the ST and RT-LSEM samples. This behavior can be associated with the higher density of crystalline defects such as dislocations and VRC (vacancy-rich clusters) introduced during the machining process. These defects provide rapid diffusion paths for solute, thus producing a high number of nucleation sites, which in turn accelerate the aging kinetics. In addition, deformation under CT could accumulate a higher concentration of dislocation and more stored energy, which provide numerous nucleation sites for the second phase and promote rapid kinetic through accelerating the nucleation events by lowering the activation energy value for the precipitation formation [21,22].

### 3.3. SEM and TEM Analysis

The SEM images of the ST, RT and CT LSEM samples are shown in Figure 6. The microstructure of ST sample mainly consists of equiaxed grains in the range of 10–40 μm with a mean grain size of 23 μm, and also subgrains of less than 5 μm can be observed. Moreover, some in-homogeneously distributed coarse constituent particles in the grain interiors and along the grain boundaries are also present in the microstructure. The results of energy dispersive spectroscopy (EDS) in Figure 7b indicate that the particle marked by spectrum1 is the Al_7_Cu_2_Fe phase, which is consistent with the findings of references [23,24]. Generally, Fe and Si are considered as impurity elements in alloys and they cannot dissolve into the matrix after solution heat treatment. They give rise to the coarse intermetallic particles, which are brittle and detrimental to most of the mechanical properties of the alloy [25]. The microstructure of CT and RT LSEM samples are similar as seen in Figure 6b,d, which are both fibrous along the extrusion direction. The grains and grain boundaries are too fuzzy to distinct in both samples.

TEM study was carried out to investigate the in-depth microstructure of material after LSEM. In this article, the samples at machining velocity of 5.4 m/min were selected as our research objects. Figure 7 illustrates the TEM images of samples prepared by RT and CT LSEM. Obviously, the degree of grain refinement for both samples is enormous, indicating that LSEM is an effective SPD method. There are significant differences between the two samples in the microstructure. The corresponding SAED (selected area electron diffraction) patterns are shown on the right of each TEM micrograph in Figure 7. It is apparent the SAED pattern of the RT-LSEM sample consists of discrete spots proving that the orientations differences between the subgrains formed during the deformation of the alloy are generally small or low angle grain boundaries (LABs). However, the presence of sharper and more continuous rings of diffraction spots in SAED pattern of the CT-LSEM sample demonstrate that the volume faction of high angle grain boundaries (HABs) increases. As the transition of LABs into HABs can be controlled by recovery rate which is accelerated with increasing temperature. At higher processing temperatures, the rate of recovery increases and therefore the dislocation annihilation becomes easier. As a result, the absorption of dislocation into LABs is limited, which means that the evolution of the microstructure into an array of HABs is more difficult [26]. During the CT-LSEM process, the accumulation of dislocations provides favorable conditions for developing HABs. 

For the CT-LSEM sample (Figure 7a), a heavily deformed microstructure with elongated grains along the shear direction is observed. The grain size are ranging from 30–100 nm in the transverse direction and 100–400 nm in the longitudinal direction respectively. Large amounts of defects tangle together forming dislocation walls and cells with diffused and non-equilibrium grain boundaries. Additionally, the wrinkles induced by dislocations prevail in the microstructure. For the RT-LSEM sample, as shown in Figure 7b, a relatively clear microstructure is demonstrated when compared with the CT-LSEM sample. Only a few elongated grains and subgrains with well-defined grain boundaries are present in the microstructure. The grain size, as well as the grain morphology, are close to each other for the two samples.

In the rest regions, however, the dislocation tangling zones decrease sharply leading to a lower dislocation density as compared with the CT-LSEM sample. Plastic work and friction between the tools and workpiece produce substantial heat during the RT-LSEM process, which facilitates the dynamic recovery and therefore the annihilation of dislocations. As dynamic recovery is mainly controlled by dislocation climb and slip, a higher temperature can trigger them and eventually lead to dislocation rearrangement and annihilation. On the other hand, deformation at CT can suppress the dynamic recovery, meaning dislocations will then be accumulated to a higher density as indicated in Figure 7a. In terms of the DSC analysis, it can be confirmed that more stored energy is kept within the CT-LSEM sample as the low temperature does not allow the release of stored energy. In other words, more crystal defects are generated when deformed at cryogenic temperature.

It should be noted that some dispersoids that have several distinct morphologies, such as rod-like, circular, and triangular shapes (marked by X, Y, and Z respectively), are seen in the RT-LSEM sample. Dispersoids, which are deliberately added for either strengthening or retardation of the recrystallization kinetics, are in the size range from 0.05 to 0.5 μm and typically contain elements such as Mg, Cu, Zn, and Cr [27]. These relatively coarse dispersodis are also referred to as the E phase in 7xxx series alloys [28]. However, the E phases can hardly be discovered in the CT-LSEM sample, which probably owes to the fact that the solutes cannot get sufficient energy to precipitate out from the matrix under very low temperatures. As a result, the Mg and Zn clusters, which are detrimental to the formability of Al alloys, are more easily formed in the RT-LSEM samples. This deduction is verified by the results that we have discussed in Section 3.1, in which the chip of CT-LSEM has better integrity than that of RT-LSEM. 

In both samples, some very fine precipitates (marked by rectangles) are present in the matrix of the alloy. As mentioned in the DSC analysis, the GP zone tends to be formed easily in the ST material after deformation even at RT. Consequently, these fine precipitates may be a mixture of GP zones and η’ phases, which are strain induced or caused by dynamic aging. The fine precipitates play a very significant role in alloys strengthening; however, the quantity is very limited. Large amounts of fine precipitates would come out from the matrix once the materials are subjected to aging treatment. The precipitation strengthening and aging behaviors will be the key points of our future research work.

### 3.4. Micro-Hardness

Figure 8 shows the micro-hardness variations of specimens as a function of the machining velocity V. The initial hardness of the ST material before LSEM is 102 HV. The hardness of the CT and RT LSEM samples at 5.4 m/min has increased to 169 HV (nearly 67% increase) and 155 HV (nearly 52% increase) respectively. The hardness of CT-LSEM samples is higher than that of RT-LSEM samples at all velocities. As can be seen from Figure 8, the microstructure of both samples is highly refined. Therefore, the large enhancement in hardness of LSEM samples could be directly be attributed to the grain refinement strengthening, one of the main strengthening mechanisms of material subjected to plastic deformation. The grain strengthening can also be explained by the Hall-Petch equation in the hardness version [29,30] instead of the flow stress version:(3)$$H_{V}=H_{0}+K_{H}d^{-1/2}$$
where $H_{V}$ is the hardness; $H_{0}$ and $K_{H}$ are the material constants; d is the mean grain size of materials. 

The dislocation strengthening, which is based on the prevention of dislocation slip and movement induced by interaction among dislocations, is another important strengthening mechanism that acts in the materials. The high density of dislocations can decrease the mean free path of mobile dislocations, thus enhancing the resistance to deformation of materials. As mentioned above, the dislocation density is higher in the CT-LSEM sample than that in the RT-LSEM sample. Accordingly, the contribution of dislocation strengthening to the hardness of the CT-LSEM alloy is more than that of the RT-LSEM alloy. 

With the increase of the machining velocity, however, the hardness value decreases for all conditions except RT-LSEM under 21.6 m/min. When the machining velocity is 10.8 m/min, the difference in hardness between CT and RT LSEM samples is 21 HV, while the value is 14 HV at 5.4 m/min. This is because of more plastic work and friction transfer into heat at 10.8 m/min, leading to a higher softening effect on the materials. Also, the hardness of the CT-LSEM samples at 5.4 m/min (169 HV) and 10.8 m/min (166 HV) is very close. As shown by Huang [11] and Magalhães [19], the grain refinement is significantly benefited by increasing the strain rate and reducing deformation temperature. The two factors are conducive to the generation and accumulation of dislocations. The strain imposed on the chips is estimated to be about 1.9 from Equation (1). The strain rates in LSEM under the machining velocities of 5.4 m/min, 10.8 m/min, and 21.6 m/min are about 3.4 × 10^3^·s^−1^, 6.5 × 10^3^·s^−1^ and 1.5 × 10^4^·s^−1^ respectively through theoretical calculations. The strain rate increases with the machining velocity while the deformation temperature almost keeps the same under CT-LSEM. Therefore, it can be deduced that the grain refinement will reach a steady state when the strain rate runs up to a specific value, regardless of the deformation temperature. 

However, the hardness of materials at 21.6 m/min is not present as expected. The hardness of RT-LSEM sample is very close to CT-LSEM sample at this velocity. It is well known that Al 7075 alloy is a precipitation hardenable alloy, in which a large number of precipitates would come out from the matrix when it is subjected to high temperature annealing. The TEM micrograph of RT-LSEM sample at 21.6 m/min is shown in Figure 9. Comparison of Figure 7 and Figure 9 suggests that more fine precipitates such as plate-shaped η’ and needle-shaped η are present and uniformly distributed in the microstructure of RT-LSEM sample at 21.6 m/min. The results also accord with our DSC analysis as discussed above. Moreover, the dislocation density in the material is further reduced. This is because the production of substantial heat induced by deformation work and friction at this machining velocity could not only trigger the precipitation behavior but also the dynamic recovery of the alloy. However, the precipitation strengthening may compensate the softening effect due to dynamic recovery, leading to a higher hardness value.

As we have discussed in Section 3.1, the chip thickness at 21.6 m/min is far less than the predefined value. Consequently, the shear strain imposed on the materials is, therefore, less than the theoretical value achieved from Equation (1). In other words, the chips at 21.6 m/min are not fully deformed as the other two conditions even though the strain rate is high enough. Generally, the grain refinement is also proportional to the shear strain within a certain range. Also, the grain size in Figure 9 is obviously greater than that in Figure 7. Hence, the grain refinement and dislocation strengthening are greatly undermined. The hardness of samples for the CT-FM remains at around 155 HV regardless of the machining velocity, which might be also attributed to the lack of constraining effect in LSEM, leading to smaller strains being imposed on the chip and therefore lower hardness values.

## 4. Summary

The machining velocity and processing temperature have great influences on the chip thickness and chip morphology. In addition, the constraining tool is the key factor that controls the chip formation process. The chip thickness decreases with the machining velocity. The chip is fully extruded and reaches the value as expected when the machining velocity is 5.4 m/min. The chip morphology of CT-LSEM is featured with better integrity than that by RT-LSEM and CT-FM. Deformation at CT can suppress the crack localization and improve the ductility and work-hardening capacity of alloys.DSC and TEM analysis reveal that many defects exist within the samples created by LSEM especially under CT, due to which the aging kinetics is accelerated. SAED analysis reveals that most of grain boundaries in RT-LSEM sample are LABs, while the volume fraction of HABs increases for the CT-LSEM sample. A higher dislocation density emerges in the CT-LSEM sample, while dynamic recovery occurs in the RT-LSEM sample owing to the heat generated through deformation, resulting in the annihilation of dislocations. The solutes can get sufficient energy to precipitate out from the matrix under CT, therefore, some coarse-grained E phases are present in the RT-LSEM sample.The hardness of chip decreases with the machining velocity for all conditions except for RT-LSEM at 21.6 m/min. On the one hand, much more heat produced at higher machining velocity softens the alloys due to the dynamic recovery. On the other hand, the hardening effect via precipitation strengthening works, which eventually compensates the softening. Additionally, the chip thickness and the shear strain cannot reach the predefined value at higher velocities, so the degree of grain refinement is not enough. The dislocation strengthening contributes more to the hardness of CT-LSEM samples.

## Figures and Tables

**Figure 1 materials-12-01656-f001:**
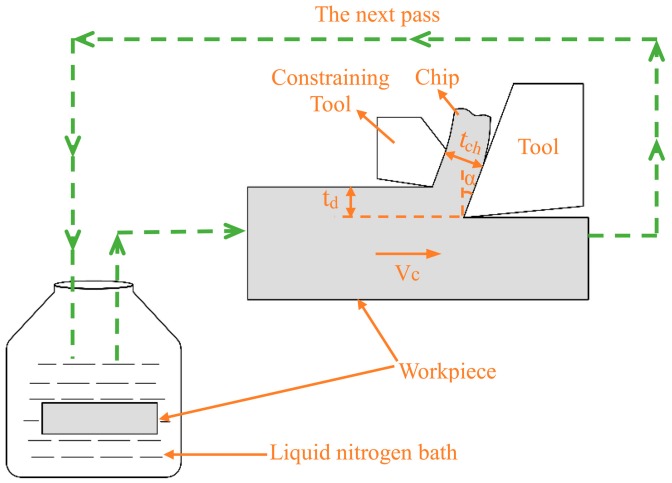
Schematic view of CT-LSEM, the workpiece should be immersed into the liquid nitrogen bath prior to machining, then CT-LSEM is conducted to obtain chips in a very short time, after that the workpiece is put back into the bath for the next pass.

**Figure 2 materials-12-01656-f002:**
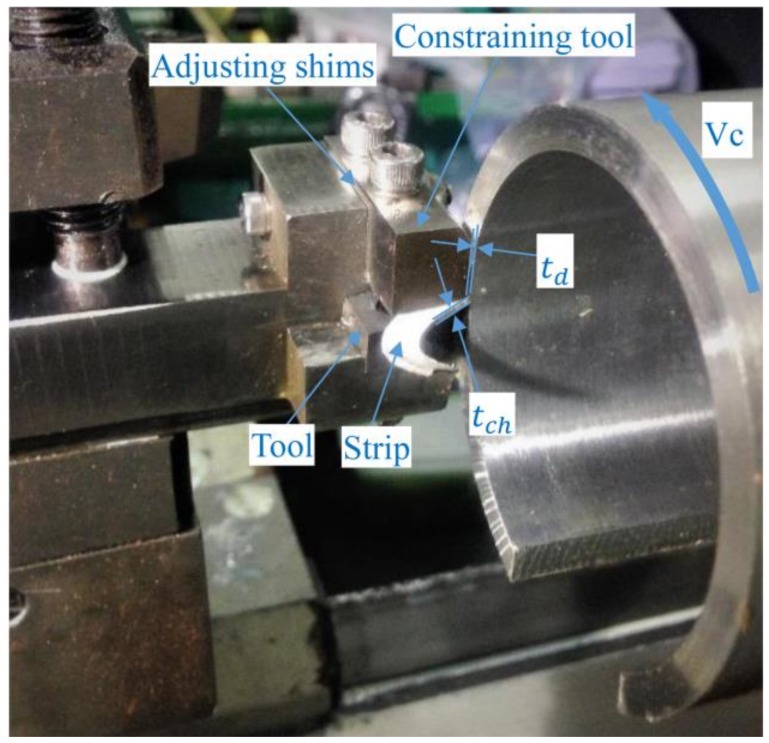
Experimental setup of LSEM and corresponding geometric parameters.

**Figure 3 materials-12-01656-f003:**
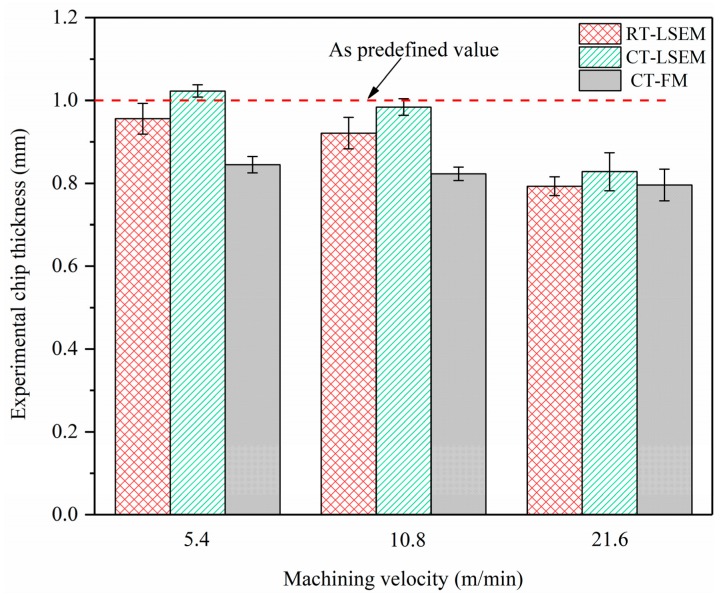
Effect of machining velocity on chip thickness of Al 7075 alloy subjected to RT-LSEM, CT-LSEM and CT-FM.

**Figure 4 materials-12-01656-f004:**
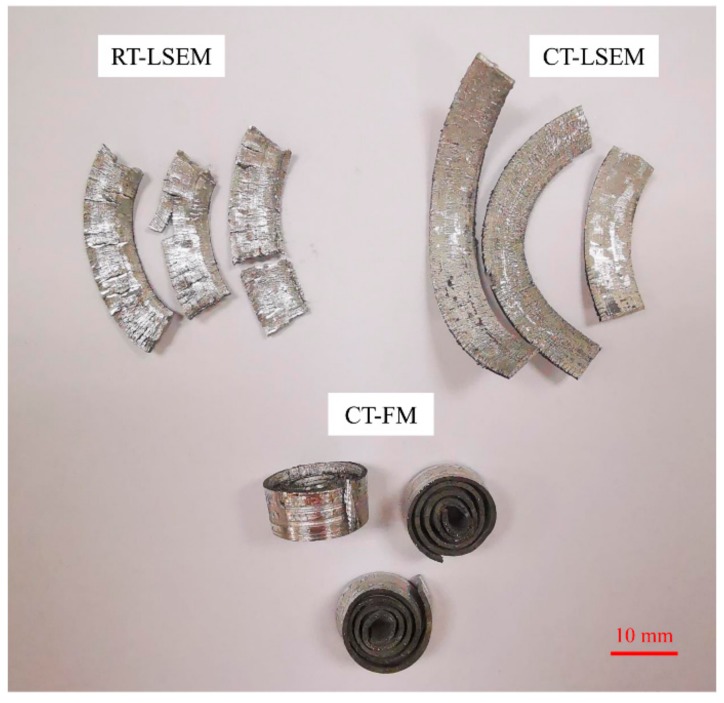
Chips and their morphologies of RT-LSEM, CT-LSEM, and CT-FM at a machining velocity of Vc = 5.4 m/min.

**Figure 5 materials-12-01656-f005:**
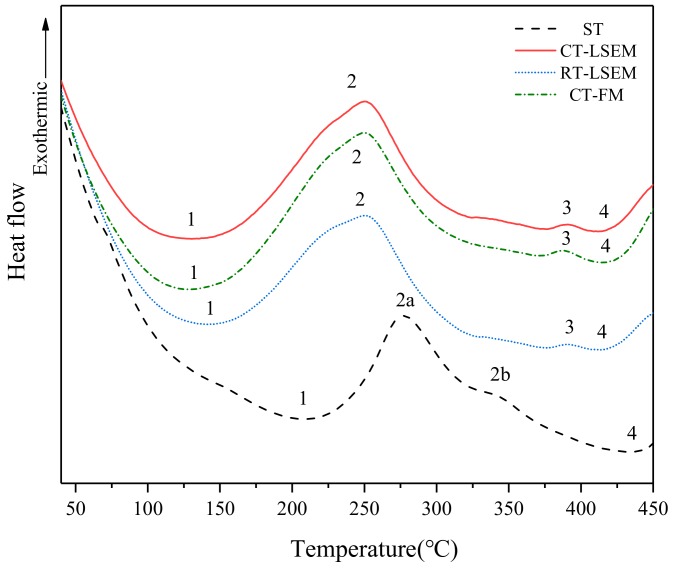
DSC scans of ST, RT and CT LSEM Al 7075 alloy samples with a heating rate of 30 °C/min at Vc = 5.4 m/min.

**Figure 6 materials-12-01656-f006:**
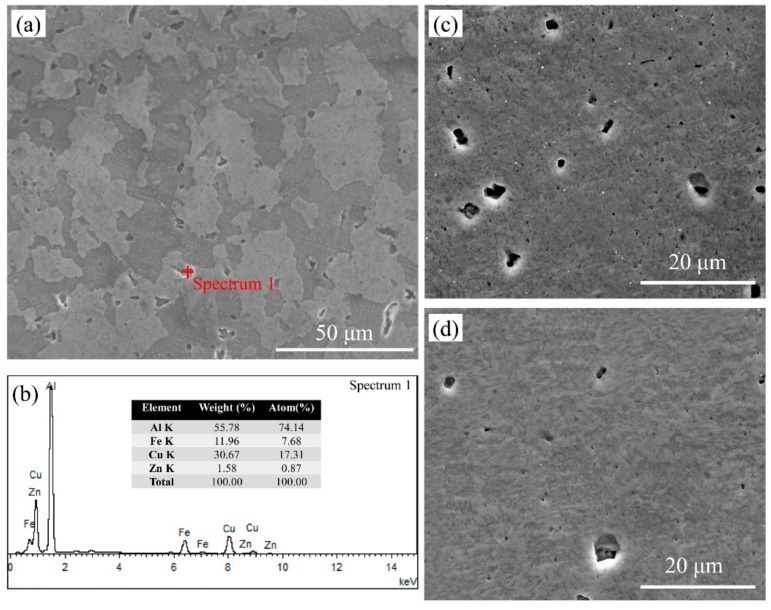
SEM micrographs of (**a**) the ST Al 7075 alloy; (**c**) the CT-LSEM sample; and (**d**) the RT-LSEM sample; (**b**) EDS results of the particle phase marked in Figure 7a.

**Figure 7 materials-12-01656-f007:**
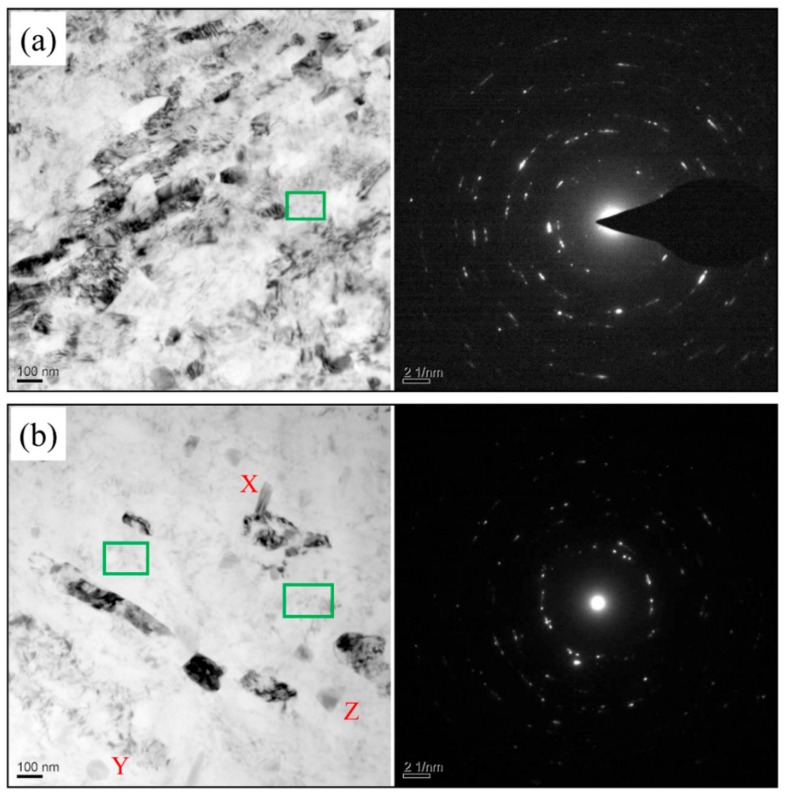
TEM microscopes and corresponding SAED patterns of Al 7075 subjected to (**a**) CT-LSEM and (**b**) RT-LSEM at a machining velocity of Vc = 5.4 m/min.

**Figure 8 materials-12-01656-f008:**
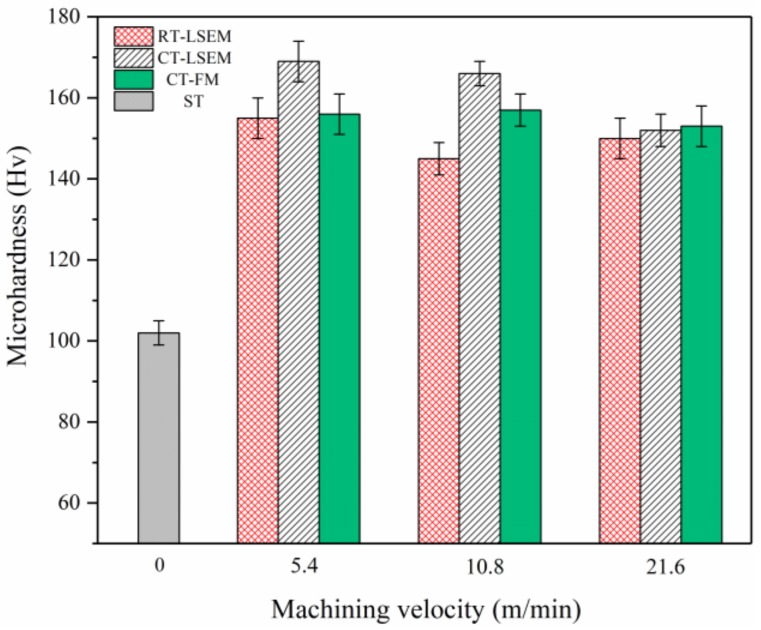
Micro-hardness of ST, RT-LSEM, CT LSEM, and CT-FM samples as a function of machining velocity, the velocity of Vc = 0 m/min represents the ST material.

**Figure 9 materials-12-01656-f009:**
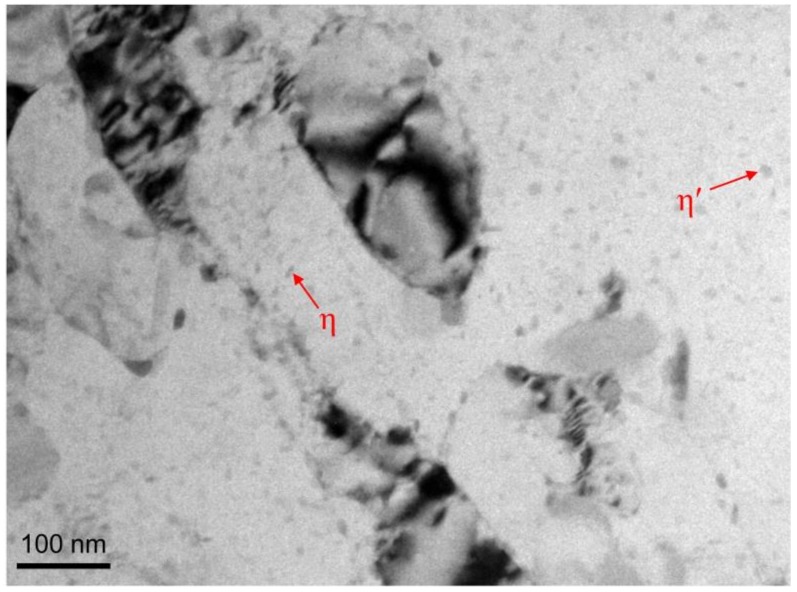
TEM microscopes Al 7075 subjected to RT-LSEM at a machining velocity of Vc = 21.6 m/min.

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
