# Peer review of "Effects of Machining Velocity on Ultra-Fine Grained Al 7075 Alloy Produced by Cryogenic Temperature Large Strain Extrusion Machining"

_materials, 2019, doi:10.3390/ma12101656_

Round 1
Reviewer 1 Report
The article "Effects of machining velocity on ultra-fine grained Al 7075 alloy produced by cryogenic temperature large strain extrusion machining" is very interesting and is very good written.
I would suggest some minor revisions to make the article more readable.
-use labels according to the ISO standard, e.g. HV, Vc, Vf
-correct the markings in Figure 1, 2, 3
-machining velocity it is determined in [m/min]
-improve the quality of the Figure 10
Although the objective is clear, how this work and finding can guide future direction are unclear.
Author Response
Response to reviews
Dear Editor and reviewer:
Thank you very much for your letter and for the reviewers’ comments concerning our manuscript entitled “Effects of machining velocity on ultra-fine grained Al 7075 alloy produced by cryogenic temperature large strain extrusion machining”. Those are professional and valuable information which have greatly helped us to revise and improve our research. We have gone through the manuscript carefully and repeatedly in these days. The amendments have been highlighted through the whole manuscript. We have also answered the reviewers’ questions and comments point by point seriously. According to the comments, we have revised the relevant part and also highlighted in the manuscript. We hope that the manuscript is now suitable publication in your journal.
Responses and revisions to comments:
The article "Effects of machining velocity on ultra-fine grained Al 7075 alloy produced by cryogenic temperature large strain extrusion machining" is very interesting and is very good written.
I would suggest some minor revisions to make the article more readable.
Thanks a lot for your scrupulous review and recognition on our manuscript. According to your advice, we have revised many relevant positions to make the article more readable.
-use labels according to the ISO standard, e.g. HV, Vc, Vf
Thanks a lot for your scrupulous review on our manuscript. Your comment is very valuable for us. The ISO standard, e.g. HV, Vc have been used in our manuscript.
-correct the markings in Figure 1, 2, 3
Thanks a lot for your scrupulous review on our manuscript. The markings in Figure 1, 2 have been corrected with new labels. According to the comment of another reviewer, Figure 2 in the initial manuscript has been removed.
-machining velocity it is determined in [m/min]
Thanks a lot for your scrupulous review on our manuscript. The unit of machining velocity has been changed to “m/min”.
-improve the quality of the Figure 10
Thanks a lot for your scrupulous review on our manuscript. The Figure 10 have been redrawn in order to improve its quality.
Although the objective is clear, how this work and finding can guide future direction are unclear.
Thanks a lot for your scrupulous review and recognition on our manuscript. Your comment is very valuable for us. We are sorry for the incomplete explanations in this part. In fact, there are many works inspired by this research are deserved to do in subsequent study. Based on your valuable comments, we have modified the manuscript as followings:
We have added these sentences “It indicates that once the appropriate processing conditions have been selected, UFG chips with better integrity, which are more suitable for practical applications, can be obtained. These findings could also provide guidance for our future work on tools optimization and practical application of chip materials” in line 143-146 and “Based on the above mentioned analysis, it can be concluded that the alloys subjected to LSEM and FM have a lot of room for improvement in strength once they are annealed with proper temperatures. This is because plenty of strengthening phases such as ηʹ and η would precipitate out from the matrix, resulting a large enhancement of strength of the materials. Therefore, the combination of CT-LSEM and subsequent aging treatment will be a focus of the forthcoming works” in line 184-189. After that, we believe that this work can well guide the future directions.
Now, we think it will be more convenient for readers to understand our experimental setup and study. Please to consider our revise.
Reviewer 2 Report
The work is of fundamental importance for the development of SPD - methods, but it is doubtful the practical application of this method in the production of any parts due to the specific shape of the samples.
1. The purpose of the work is not precisely formulated.
2. There is no reference to Figure 2, this figure largely repeats the scheme shown in Figure 1. The only difference is in the arrows of the direction of the impact velocity, so it can be removed.
3. What is the shape and size of the sample?
4. The text in 64-74 duplicates the information in paragraph 2. I think, instead, here it is necessary to formulate the purpose and objectives in more detail.
5. Is it possible to estimate the strain rates corresponding to the machining velocities given?
Author Response
Response to reviews
Dear Editor and reviewer:
Thank you very much for your letter and for the reviewers’ comments concerning our manuscript entitled “Effects of machining velocity on ultra-fine grained Al 7075 alloy produced by cryogenic temperature large strain extrusion machining”. Those are professional and valuable information which have greatly helped us to revise and improve our research. We have gone through the manuscript carefully and repeatedly in these days. The amendments have been highlighted through the whole manuscript. We have also answered the reviewers’ questions and comments point by point seriously. According to the comments, we have revised the relevant part and also highlighted in the manuscript. We hope that the manuscript is now suitable publication in your journal.
Responses and revisions to comments:
The work is of fundamental importance for the development of SPD - methods, but it is doubtful the practical application of this method in the production of any parts due to the specific shape of the samples.
Thanks a lot for your scrupulous review and recognition on our manuscript. Your comments is very valuable for us. In fact, though materials prepared by LSEM exhibit excellent mechanical properties, the shape size of them is the largest constriction that limits their practical application. However, the research works related to scaling-up of the bulk UFG samples through LSEM are also in our schedule.
In this article, an innovative SPD method, which combines cryogenic technique and LSEM, i.e., CT-LSEM, was put forward. This is a useful attempt to explore new strategies in SPD field, in other words, we are more inclined to foundational research. We are mainly focus on studying how the CT exerts influences on the mechanical properties and microstructure evolutions of the alloy during LSEM. Also, the comparative effects between convention machining and CT-LSEM are deserved to be studied.
In order to facilitate the preliminary study, the structure and size of LSEM tools should be limited to a reasonable scope, which results in the smaller chip size as indicated in figure. 4 in the manuscript. We have been working on the practical application of chips including magnifying the size of bulk UFG materials. In fact, we just need to adjust the tool structure accordingly, then the chip size will be enlarged which can satisfy the practical application.
The chip samples in this research also have potential application values in some fields such as Microelectro Mechanical Systems (MEMS) components. Figure. 1 shows the small scale gears produced from the LSEM chips using micro-EDM (doi:10.1016/j.msea.2008.02.056). The chip samples can also be milled to powders using ball milling as shown in figure 2 (doi.org/10.1016/j.powtec.2013.07.028). These powders are then consolidated with nano particles to produce nano-composite materials.
Figure. 1 Micro gears produced from LSEM chips(please refer to the word version)
Figure. 2 Chips morphology and milled powders (please refer to the word version)
1. The purpose of the work is not precisely formulated.
Thank you for your scrupulous review on this issue. We are sorry for the incomplete description of purpose in this work. After careful reorganization and inspection of this research, we have modified the manuscript as followings:
We have added the sentences “The LSEM process could impart large strains to the materials in a single step of deformation. Comparing with the conventional SPD methods, LSEM is more efficient to fabricate UFG materials.” in line 56-58 to explain the advantages of LSEM one more step.
We have added the sentences “ The Al 7075 was ST in order to obtain a supersaturated solid solution (SSS) due to its higher work hardening rate and better microstructure refinement” in line 68 to explain the reason for ST.
We have changed the sentences “ Room temperature large strain extrusion machining (RT-LSEM) was also conducted for comparison” to “ Room temperature large strain extrusion machining (RT-LSEM) and free machining (FM) under CT (CT-FM) were also conducted for comparison”.
We have deleted and revised the redundant descriptions: “Microstructure and mechanical properties of the alloy were characterized by employing differential scanning calorimetry (DSC), scanning electron microscopy (SEM), transmission electron microscopy (TEM), and Vickers hardness measurements. For the RT and CT LSEM chips, the effects of velocity on chip thickness and morphology, as well as the microstructure and mechanical properties, were studied in detail. The results demonstrate that CT-LSEM has advantages over RT-LSEM and CT-FM in improving the mechanical properties and ensuring the integrity of chips when choosing a proper machining velocity. Therefore, the present work mainly focuses on developing a new method to prepare UFG alloys and identifying their differences in microstructure evolution and mechanical properties”, and changed them to “ The machining velocity plays a very significant role in the chip formation process and is the main factor that influences the heat generation in the deformation zones. As the heat induced by deformation and friction is detrimental to the mechanical properties owing to some microstructure changes within the material, it is therefore necessary to reduce the heat effects on material during machining. Additionally, the chips prepared by different methods might present distinctive morphologies. As far as we know, CT-LSEM with the starting materials in the ST state under various machining velocities has not been reported for precipitation hardenable alloy (Al 7075) in opened literatures. Since the strengthening mechanisms of 7xxx series alloy mainly include grain refinement, dislocation, and precipitation, their contributions to alloys subjected to RT and CT LSEM, as well as CT-FM are very worthy of research. Therefore, the overall purpose of present work mainly focuses on: (i) developing a new method to prepare UFG alloys; (ii) studying the effects of machining velocity on the chip formation; (iii) identifying the differences in microstructure evolution and mechanical properties; (iv) investigating the strengthening mechanisms of chip samples. The microstructural and mechanical features were examined by employing differential scanning calorimetry (DSC), scanning electron microscopy (SEM), transmission electron microscopy (TEM), and Vickers hardness measurements”, which further clarify the purpose of the article. After revising, the objectives of this work are believed to be accurately formulated.
Now, we think it will be more convenient for readers to understand our study. Please to consider our revise.
2. There is no reference to Figure 2, this figure largely repeats the scheme shown in Figure 1. The only difference is in the arrows of the direction of the impact velocity, so it can be removed.
Thanks a lot for your scrupulous review on our manuscript. Your comment is very valuable for us. Indeed, the definition of CT-FM is very easy to be understand in terms of Figure 1. The existence of Figure 2 in the manuscript is not necessary. Therefore, we have removed it from the revised version. Please to consider our revise.
3. What is the shape and size of the sample?
Thanks a lot for your scrupulous review on our manuscript. Generally, the chips are featured with strip shaped. The chip length, width and thickness are in the range of 30-400 mm, 5-10mm, and 0.4-1.5 mm respectively. In fact, all sizes can be adjusted to a proper values as long as needed through altering the structure of the combined tool as mentioned above. In this article, the chips prepared by LSEM are strip shaped with 30-80 mm long, 6mm width, and ~1mm thick while that by CT-FM are curl shaped with similar width and thickness values with the LSEM samples. This is also the consideration for the convenience of research.
4. The text in 64-74 duplicates the information in paragraph 2. I think, instead, here it is necessary to formulate the purpose and objectives in more detail.
Thank you for your scrupulous review and pertinent comment on this issue. Your comment is very valuable for us. The text in 64-74 is really redundant and it has been significantly revised according to your comment. The purpose and objectives of this article are formulated in more detail in paragraph 2. Please refer to our answers to comment 2, in which the specific modifications can be obtained.
5. Is it possible to estimate the strain rates corresponding to the machining velocities given?
Thanks a lot for your scrupulous review and pertinent comment on our manuscript. As we have mentioned in the manuscript that the strain rate greatly influences the grain refinement of the alloys. Theoretically, the schematic of shear deformation in LSEM is illustrated in Fig. 3. When the parallelogram OHNM is subjected to shear deformation, it turns into the parallelogram OGPM. Then the strain rate in cutting is:
where V is the cutting speed, is the time elapsed for the metal to travel a distance along the shear plane, and is the thickness of the shear zone. A reasonable mean value for the spacing of successive slip plane () would appear to be about 25 μm (M.C. Shaw. Metal cutting principles. Clarendon press. Oxford. 1984.). It can thus be calculated from the above equation that the strain rates for LSEM under 90 mm/s (5.4 m/min), 180 mm/s (10.8 m/min), and 390 mm/s (21.6 m/min) are about 3.4×103 s-1, 6.5×103 s-1, and 1.5×104 s-1 respectively.
Ps. the equation applies equally to the free machining (FM) process.
Fig. 3 Schematic diagrams of shear deformation for LSEM. (please refer to the word version)
In addition to theoretical calculations, we also use the FEM method to predict the strain, strain rate, temperature, and cutting fore etc. in the machining process, which is proved to be more efficient and convenient.
In this article, the analysis of strain rate is not our focus, therefore we just mention it and do not study it in detail. Based on your valuable comments, we have modified the manuscript as followings:
We have added the sentences “The strain imposed on the chips is estimated to be about 1.9 from Eq. (1). The strain rates in LSEM under the machining velocities of 5.4 m/min, 10.8 m/min, and 21.6 m/min are about 3.4×103 s-1, 6.5×103 s-1, and 1.5×104 s-1 respectively through theoretical calculation” in line 310-313.
Now, we think it will be more convenient for readers to understand our study. Please to consider our revise.

Round 2
Reviewer 2 Report
The authors corrected all comments, I think that the article requires acceptance for publication.